# Vectorized Mathematical Model of a Slip-Ring Induction Motor †

**Miroslaw Wcislik** [1,*] **, Karol Suchenia** [2] **and Andrzej Cyganik** [3]

1   Faculty of Electrical Engineering, Automatic Control and Computer Science, Kielce University of Technology, 25-314 Kielce, Poland
2   Faculty of Electrical and Computer Engineering, Cracow University of Technology, 31-155 Cracow, Poland; karol.suchenia@pk.edu.pl
3   Siemens Sp z o. o, RC-PL DF FA, 30-443 Kraków, Poland; andrzej.cyganik@siemens.com
*   Correspondence: m.wcislik@tu.kielce.pl; Tel.: +48-41-342-4212
†   This paper is an extended version of our paper published in 19th International Symposium on Electromagnetic Fields in Mechatronics, Electrical and Electronic Engineering (ISEF), Nancy, France, 29–31 August 2019, added to IEEE Xplore in 20 May 2020.

**Abstract:** The paper deals with the modeling of a slip-ring induction motor. Induction motors are very often used in industry and their suitable model is needed to reduce control and operating costs. The identification process of self and mutual inductances of the stator and rotor, and mutual inductances between them in the function of the rotor rotation angle is presented. The dependence of each inductance on the rotor rotation angle is determined experimentally. The inductance matrix is then formulated. Taking the magnetic energy of the inductances and kinetic energy of the rotor into account, the Lagrange function is defined. Next, the motor motion equations are obtained. After making some algebraic transformations and using the dimensionless variables, the motion equations of electric circuits and of the mechanical equation are written separately in the forms facilitating their solution. The solution was obtained using the Simulink model for the stator and rotor currents in the form of vectors. The simulation was controlled by MATLAB script. The results of the simulation are presented in the form of basic variables time courses and compared with some values calculated with the use Steinmetz model of induction motor. The work is followed by two appendices, which contain procedures for determining the inverted inductance matrix.

**Keywords:** slip-ring induction motors; modeling; inductance matrix; motion equation; simulation

## 1. Introduction

AC induction motors definitely have advantage over DC ones. They are more reliable and their purchase and maintenance costs are significantly lower [1]. This is mainly due to a simple design of the rotor and that the stator is the only element which is connected to a power supply, as well as due to economies of scale. The popularity of induction motors is confirmed by the data published in [2], where it was stated that induction motors constitute 95% of driving devices and consume up to 40–50% of the total produced electricity.

Furthermore, AC induction motors can be used in difficult operating conditions, including areas with high levels of dust, chemically aggressive atmosphere and even in explosion hazard zones as speed control and positioning systems. That is why the induction motors are commonly used in different types of industrial drive systems with velocity or position as controlled variable. It is expected that in the next decade up to 50% of all electric motors will contain induction motors powered from power electronic systems [3].

The driving torque of the induction motor results from the stator and rotor interactions. There are two types of rotors: caged and wound [2–4]. The design and construction of a cage rotor is simpler, but its mathematical description is more complex than a wound rotor [5]. A wound rotor has windings connected in a star. The connection point of these windings is usually isolated. The other ends of these windings are led to the rings, on which the graphite brushes slide. External resistors can be connected through these brushes. These resistors can facilitate starting of the motor and shape its operating characteristics [3]. The heat generated in the external resistors does not directly affect the motor interior temperature. As a result, the insulation of the windings and bearings age more slowly.

Slip-ring induction motors are used when high starting torque or low starting current is required. They are particularly suitable for driving systems of the machines with high inertia loads. Currently, ring motors with powers up to 20 MW are produced.

The use of induction motors with slip rings with a wounded rotor circuit connected to a variable external resistance allows speed regulation in a considerable range. However, the thermal losses of resistors associated with low speed motor operation are a serious problem. Slip ring motor drives are often using systems to recover electricity from the rotor circuit, which is rectified and returned to the power supply by means of a variable frequency drive [6].

An alternating current power system, developed by George Westinghouse, was introduced as electric power transmission system in the late 1880s and early 1890s. The first AC commutator-free induction motors were independently invented by Galileo Ferraris and Nikola Tesla, respectively in 1885 and in 1887. The three-phase circuit was first applied by Michal Doliwo-Dobrowolski in 1889. First mathematical model of the induction motor was published in 1897 by Charles Steinmetz. He proposed T-equivalent circuit model. The model is a single-phase circuit of a multiphase induction motor. On the basis of Steinmetz equivalent circuit analysis, it is possible to determine many useful relationships between circuit parameters, current, voltage, speed, power and torque. They describe how electrical input variables are transferred into mechanical output in induction motor. The Steinmetz diagram has been widely used in electrical engineering in the unchanged form for over a hundred years. In [6,7] the author uses the Steinmetz diagram, also for three-phase induction motors. The elements of the single-phase equivalent diagram are recalculated from the given parameters of three-phase circuit of the induction motor.

Some extension of the Steinmetz diagram was accomplished in [8,9] by introducing an ideal rotating transformer (IRTF) between magnetizing inductance and load resistance. The use of this element facilitated the modeling of electric machines. In the chapter on inductive machines, a universal model of winding stream connections was introduced, which enables the transformation to a two-phase circuit in frame d-q-0 that leads to a simplified machine model with IRTF. This model is the basis for a universal model of a magnetic field-oriented machine, which enables analysis and facilitates understanding of the dynamics of inductive machines. This model is the basis for the development of field-oriented control [8].

Futhermore, in [4], chapter 3 considers the induction motor scheme developed by Steinmetz. In this diagram, the resistance of losses in iron is additionally introduced in parallel to magnetization inductance and the resistance of rotor windings is distinguished. The analysis of different types of working areas and working characteristics in these areas was carried out. Chapter 4 presents, between others, a three-phase model of an induction motor and the equations of magnetic fluxes and inductance matrixes. Next, the Park transformation was applied, and four scalar differential equations of ordinary electrical part were obtained for coordinates d-q-n (equivalent d-q-0). Schemes in Simulink for scalar variables solving differential equations and sample diagrams of dynamic processes in the induction motor are also presented.

Summarizing these publications review it can be concluded that the T-equivalent model is the most commonly used one. The element of this model, which is defined as the quotient of resistance and slip, it is a non-linear circuit part. It should be stressed that according to [6] the Steinmetz model is single phased and valid only in steady-state balanced circuit condition. However in drive systems

there is often an alternating mechanical load and a three-phase power supply is used, which is not always symmetrical. In addition, the asymmetry of stator and rotor circuits and the non-sinusoidality of currents are associated with the creation of a potential difference between the central points of the winding stars and supply voltages or load resistance. This phenomenon is not taken into account in the publications in question. Especially since the transition from three-phase variables to orthogonal coordinates d-q-0 is associated with a common reference point.

The magnetic streams coupling the stator and rotor windings flow twice through the air gap. Hence it can be assumed that the magnetic fluxes in the induction motor are proportional to the currents, and the stator and rotor currents should be assumed as state variables, respectively, and so applied in [5,10,11]. However, in a three-phase system without a neutral conductor there are only two independent currents. It means that the state equations of the stator and rotor circuits should be of second order. However, the state coordinates of these circuits need not be orthogonal. It can be two of three phase currents.

The purpose of the analysis is to determine the mathematical model of the slip ring induction motor. This model should make it possible to analyze both the influence of motor parameters and power quality disturbances occurring in the motor supply circuit on the output torque of the motor and the impact of dynamic load moments on the motor shaft on its supply system. The models presented above are not sufficient for these purposes

The basis of the induction motor model is the inductance matrix. The form of this matrix is defined in [4,5,10,12]. The first chapter proposes a method of measuring the elements of this matrix and their values are determined. Direct measurements of the inductance matrix elements and checking its structure can be done only for the slip ring induction motor. Therefore, the paper includes an analysis of the slip ring induction motor. In practice, the model of this motor is often used as a cage induction motor model [3].

Taking into account the above remarks, the equations of the state of the electric part of the motor circuits have been recorded in the vector-matrix form of the 6th order and then transformed into the 4th order. Non-dimensional variables resulting from the equations and time scaling were used. The results of model simulation with dimensionless variables were converted into physical variables. The simulation was conducted for a balanced motor. Thanks to that, it was possible to use the Steinmetz model to verify the obtained results.

An original procedure for inverting the inductance matrix was developed. It allowed to present the equations in a form facilitating the solution of model equations in MATLAB-Simulink system. The operating diagram of the analyzed model is much smaller and simpler than the one presented in [4,13] and allows to conduct simulation experiments to study power quality disturbances and dynamic mechanical loads of the motor.

## 2. Inductance Matrix of a Slip-Ring Induction Motor

The three-phase circuits of both the stator and rotor should be considered as part of the motor dynamics analysis. In each phase of each circuit there are windings, which are mutually coupled. The type of couplings depends on the rotor rotation angle in relation to the stator. The couplings between the windings also depend on the properties of the magnetic circuit. In the induction motor, the air gap between the rotor and the stator is important. The magnetic flux generated by the currents of the stator flows through the gap twice. This allows to assume that the motor magnetic circuit is linear and may be described using inductances. That is why the base of the induction motor mathematical model is a matrix of inductances. The matrix may be described experimentally. The separated physical parts of the motor are powered and relevant elements of the inductance and resistance matrix are calculated. One drawback of this approach is that the measurements can be performed only on a real, existing machine. The results of these measurements allow us to determine which parameters are relevant for a given model. In the process of identifying the parameters of the model, the measurements are performed as the function of the rotor rotation angle.

The inductance matrix **L** of an induction motor depends on rotor rotation angle and consists of four submatrices 3 × 3: **Ls**—the stator matrix of self-and mutual inductances, **Msr**—the matrix of mutual inductances of the rotor in relation to the stator, **Lr**—the rotor matrix of self—and mutual inductances and **Mrs**—the matrix of mutual inductances of the stator in relation to the rotor.

$$\mathbf{L}(\varphi) = \begin{bmatrix} \mathbf{Ls} & \mathbf{Mrs} \\ \mathbf{Msr} & \mathbf{Lr} \end{bmatrix}, \tag{1}$$

Taking into account the matrix **L** symmetry [3], it can have 21 different elements. Simultaneous identification of all these parameters causes the parameter measuring system to be rather complex. Therefore, the measurements were carried out in several stages. One of the stator or rotor windings was connected to a 50 Hz AC source and the source current and the voltages of all motor windings were measured. Measurements of both the powered winding and the remaining ones were performed using an eight-canal simultaneous measurement system. The diagram of connections for the case when one winding of the stator is powered is presented in Figure 1. The supplied winding was marked with the thick line.

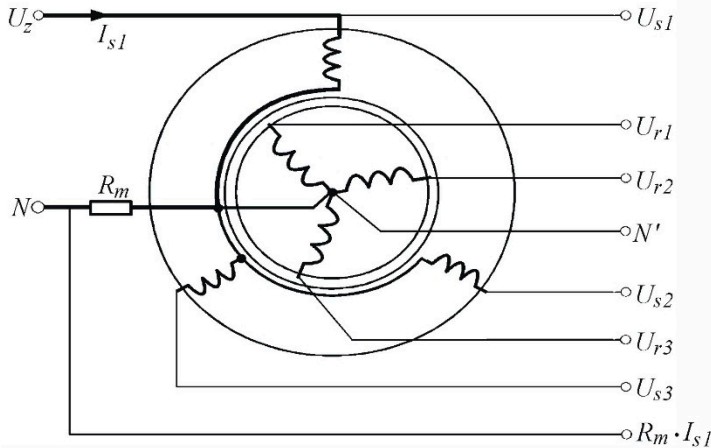

**Figure 1.** The diagram of connections for one powered winding of the stator.

In the first stage of parameters determination, the supply was connected to the first winding of the stator and both the supply current and voltages: $I_{s1}$, $U_{s1}$, $U_{s2}$ and $U_{s3}$ were measured. The measured voltages are described by equations:

$$L_{s1}\frac{dI_{s1}}{dt} + R_{s1} \cdot I_{s1} = U_{s1}, \tag{2}$$

$$L_{sm12}\frac{dI_{s1}}{dt} = U_{s2}, \tag{3}$$

$$L_{sm13}\frac{dI_{s1}}{dt} = U_{s3}, \tag{4}$$

After using the Golay-Sawitzki filter to the currents and voltages signals, the inductances and resistances were determined from the above-mentioned equations with the use of the least squares' method. To determine mutual inductances, it is necessary to use the measurements taken at the supply of chosen one winding—Figure 1.

The $L_{s1}$ inductance of the winding 1 of the stator is sum of leakage inductance $L_{s\_1}$ and magnetizing inductance $L_{sm1}$:

$$L_{s1} = L_{s\_1} + L_{sm1}, \tag{5}$$

The magnetizing inductance $L_{sm1}$ of the winding 1 equals the negative sum of mutual inductances of the remaining windings $L_{sm12}$ and $L_{sm13}$:

$$L_{sm1} = -(L_{sm12} + L_{sm13}),\tag{6}$$

The measurements were performed winch induction motor of SZUDe36a 2P44 type. It is the three-phase slip-ring and has three pairs of poles. The nominal stator current for the star connection of windings was equal to 4.1 A, the nominal rotor current for the star connection of windings was 21 A and the nominal motor speed was 920 revolutions per minute. The rated power is 1.5 kW. The measurements were executed at the supply of the windings with the current equal to approx. 0.25 of the rated current of given winding.

All stator windings were measured. The averaged measurement results showed that the stator magnetizing inductance was equal to $L_{sm} = 0.187$ H and the stator leakage inductance to $L_{s\_} = 0.0293$ H. The average phase resistance of windings of the stator was also measured. It amounted to $R_s = 10.5\ \Omega$ and contained the eddy currents resistance and the windings copper resistance equaling $R_{sDC} = 3.7\ \Omega$, (measured with the multimeter).

The measured values of self and mutual inductance coefficients of stator phase windings were very close. The parameters differences for the circuit of different phases in the function of the rotation angle did not exceed 2% of their mean value. Therefore, it was acknowledged that they were equal for all phases of the motor and independent on the rotation angle [4,5]. This applies also to both the leakage and the magnetizing inductances. It means that the matrix **Ls** is symmetric and the elements of the diagonal are equal and amount to $L_s$. The elements outside the diagonal equal $-0.5L_{sm}$:

$$\mathbf{Ls} = \begin{bmatrix} L_s & -\frac{1}{2}L_{sm} & -\frac{1}{2}L_{sm} \\ -\frac{1}{2}L_{sm} & L_s & -\frac{1}{2}L_{sm} \\ -\frac{1}{2}L_{sm} & -\frac{1}{2}L_{sm} & L_s \end{bmatrix} = L_{sm} \cdot \begin{bmatrix} 1+\sigma_s & -1/2 & -1/2 \\ -1/2 & 1+\sigma_s & -1/2 \\ -1/2 & -1/2 & 1+\sigma_s \end{bmatrix} = L_{sm} \cdot \mathbf{mLs},\tag{7}$$

where:

$$\sigma_s = \frac{L_{s\_}}{L_{sm}} = \frac{L_s - L_{sm}}{L_{sm}}$$

$$\mathbf{Ls} = L_{sm} \cdot [\,\mathbf{1m} + \sigma_s \cdot \mathbf{1\_}\,] = L_{sm} \cdot \mathbf{mLs},\tag{8}$$

where:

$$\sigma_s = L_{s\_}/L_{sm}; \quad \mathbf{1m} = \begin{bmatrix} 1 & -0.5 & -0.5 \\ -0.5 & 1 & -0.5 \\ -0.5 & -0.5 & 1 \end{bmatrix}; \quad \mathbf{1\_} = \begin{bmatrix} 1 & 0 & 0 \\ 0 & 1 & 0 \\ 0 & 0 & 1 \end{bmatrix},$$

Similar measurements were conducted for the rotor. It was obtained the averaged magnetizing inductance $L_{rm} = 3.9$ mH and the leakage inductance $L_{r\_} = 0.55$ mH from (5) and (6). The average phase resistance of the stator windings amounted to $R_r = 0.523\ \Omega$ and contained the resistance introduced by eddy currents and the windings (copper) resistance equaling $R_{rDC} = 0.2\ \Omega$.

The matrix **Lr** has the form similar to that of **Ls** matrix. The elements of the diagonal are equal and amount to $L_r$. and the elements outside the diagonal are equal to—$0.5L_{rm}$.

$$\mathbf{Lr} = \begin{bmatrix} L_r & -\frac{1}{2}L_{rm} & -\frac{1}{2}L_{rm} \\ -\frac{1}{2}L_{rm} & L_r & -\frac{1}{2}L_{rm} \\ -\frac{1}{2}L_{rm} & -\frac{1}{2}L_{rm} & L_r \end{bmatrix} = L_{rm} \cdot \begin{bmatrix} 1+\sigma_r & -1/2 & -1/2 \\ -1/2 & 1+\sigma_r & -1/2 \\ -1/2 & -1/2 & 1+\sigma_r \end{bmatrix} = L_{rm} \cdot \mathbf{mLr},\tag{9}$$

where:

$$\sigma_r = \frac{L_{r\_}}{L_{rm}} = \frac{L_r - L_{rm}}{L_{rm}},$$

The mutual inductances between the stator and rotor as well as between the rotor and stator in the function of the rotor rotation angle were determined for the supply given to one winding of the

stator or rotor, respectively. In case of the supply given to the rotor it is assumed that the voltages on stator windings are described by the equations:

$$M_{rsn} \cdot \frac{dI_{r1}}{dt} = U_{sn}, \text{ where } n = 1, 2, 3 \tag{10}$$

.

As in the previous case the signals of currents and voltages were filtered. The elements of the matrix of mutual inductances in the function of the rotor rotation angle were identified using the least squares method. The measurements were carried out for the supply given to the individual windings of the rotor.

The waveforms of coefficients of the mutual inductances of stator windings in relation to the rotor first phase winding versus the rotor rotation angle are presented in Figure 2.

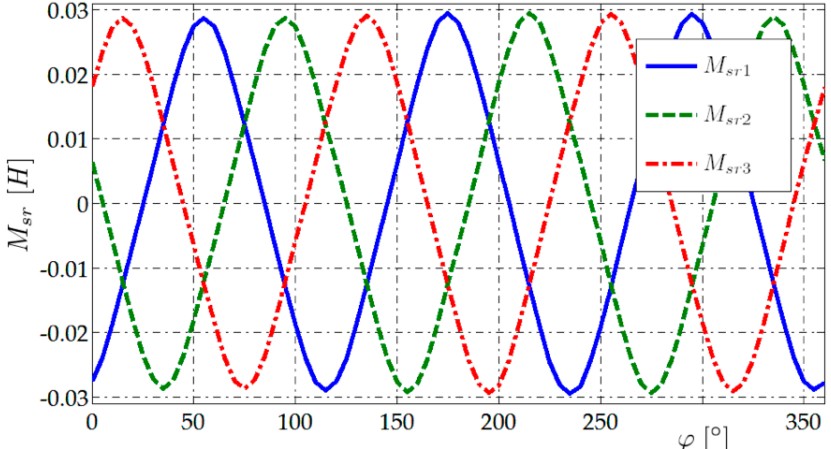

**Figure 2.** The coefficients of the mutual inductance of the stator windings in relation to the first phase rotor winding versus the rotor rotation angle.

The coefficients of mutual inductances between stator and rotor are periodic functions of the rotor rotation angle. Their period amounts to 120 degrees and results from the number of poles pairs. Their amplitudes differ at most by approx. 2%. The inductances generate the cyclic matrix-circulant [14]:

$$\mathbf{Mrs} = M_{rs} \cdot \mathbf{mC}, \tag{11}$$

where:

$$\mathbf{mC} = \begin{bmatrix} \cos(3\varphi) & \cos(3\varphi + q) & \cos(3\varphi + 2 \cdot q) \\ \cos(3\varphi + 2 \cdot q) & \cos(3\varphi) & \cos(3\varphi + q) \\ \cos(3\varphi + q) & \cos(3\varphi + 2 \cdot q) & \cos(3\varphi) \end{bmatrix}, \quad q = 2 \cdot \pi/3, \tag{12}$$

In the next stage the stator winding was supplied and measurements of voltages on rotor windings in the function of the rotation angle were conducted. The voltages are described by an equation analogous to (10). The measured coefficients are also periodic functions of the rotor rotation angle they generate the circulant as well. Their amplitudes differ by 2%.

$$\mathbf{Msr} = M_{sr} \cdot \mathbf{mC}^{\mathbf{T}}, \tag{13}$$

The amplitude of the mutual inductances among the stator and rotor equals $M_{sr}$ = 0.0275 H. The amplitude of the mutual inductances between the rotor and stator is approx. 4% smaller and amounts to $M_{rm}$ = 0.0264 H.

In the following part of the chapter it is assumed that the $M_{rs}$ and $M_{sr}$ inductances are equal to:

$$M = \sqrt{L_{sm} \cdot L_{rm}} = 0.027 \text{ H},\tag{14}$$

with the accuracy of about 2% of $M_{rs}$ and $M_{sr}$ values.

Basing upon the measurements we have concluded that phase resistances of the stator circuits have very close values. This applies also to the rotor resistances per phase. The phase resistances of the stator and rotor circuits are placed on diagonals of the matrices respectively **Rs** and **Rr**.

Finally, the inductance matrix of the motor is assumed in the form:

$$\mathbf{L}(\varphi) = \begin{bmatrix} L_{sm} \cdot \mathbf{mLs} & M \cdot \mathbf{mC}(\varphi) \\ M \cdot \mathbf{mC}(\varphi)^T & L_{rm} \cdot \mathbf{mLr} \end{bmatrix},\tag{15}$$

The resistances matrix of the motor is also a concatenation of the stator and rotor diagonal resistances matrices:

$$\mathbf{R} = \begin{bmatrix} \mathbf{Rs} & 0_{3x3} \\ 0_{3x3} & \mathbf{Rr} \end{bmatrix},\tag{16}$$

The $0_{nxn}$ denotes the zero matrix of n x n dimensions. The phase resistances of the stator and rotor circuits are placed on diagonals of the matrices:

$$\mathbf{Rs} = \begin{bmatrix} R_{s1} & 0 & 0 \\ 0 & R_{s2} & 0 \\ 0 & 0 & R_{s3} \end{bmatrix}, \mathbf{Rr} = \begin{bmatrix} R_{r1} & 0 & 0 \\ 0 & R_{r2} & 0 \\ 0 & 0 & R_{r3} \end{bmatrix},\tag{17}$$

In general, in the model the stator phase winding resistances can have different values. The same can be for the rotor winding. However, basing upon the measurements, it was assumed in simulation that phase resistances of the stator circuits are equal. This applies also to the rotor resistances per phase.

The inductance matrix of a slip-ring motor (1) depends on rotor rotation angle and together with motor currents (time derivative of electric charges) determine the magnetic energy, which aggregated with the rotor kinetic energy form the Lagrange function.

$$fL = \frac{1}{2}\mathbf{I}^T \cdot \mathbf{L}(\varphi) \cdot \mathbf{I} + \frac{1}{2}J \cdot \omega^2,\tag{18}$$

where:

$\mathbf{L}(\varphi)$—the matrix of inductances, dependent on the angle of rotor rotation in relation to a stator,

$\mathbf{I} = \dot{\mathbf{Q}}$—the motor currents column vector,

$\omega = \dot{\varphi}$—the angular velocity of the rotor,

$J$—the moment of rotor inertia.

The Lagrange function is the difference of kinetic and potential energies and it does not take into account the friction and external forces. Consequently, the Lagrange function does not describe the energy flow [15].

Basing on the Lagrange function the d'Alembert-Lagrange equation may be formulated [15]. Using the equation with virtual velocities as variations of virtual coordinates it may be easy checked that induction motors are holonomic systems. Thanks to that, after using the Euler-Lagrange equations and taking into account forces of friction and external excitation, the motion equations may be obtained in the form:

$$\frac{d}{dt}\frac{\partial fL}{\partial \dot{\mathbf{Q}}} = \frac{d}{dt}\frac{\partial fL}{\partial \mathbf{I}} = \mathbf{Uz} - \mathbf{R} \cdot \mathbf{I} - \mathbf{Uo},\tag{19}$$

$$\frac{d}{dt}\frac{\partial fL}{\partial \dot{\varphi}} - \frac{\partial fL}{\partial \varphi} = T_L - T_F(\dot{\varphi}),\tag{20}$$

where **Uz**—denotes the supply voltage vector, **Uo**—the vector of voltages between s neutral points, R—the windings resistance matrix, $T_L$—the mechanical load torque, $T_F$—the mechanical friction torque

After substitution of Lagrange function into the Equations (19) and (20) and some transformations we get

$$\mathbf{L}(\varphi)\frac{d\mathbf{I}}{dt} + \frac{\partial \mathbf{L}(\varphi)}{\partial \varphi} \cdot \frac{d\,\varphi}{d\,t} \cdot \mathbf{I} + \mathbf{R} \cdot \mathbf{I} + \mathbf{Uo} = \mathbf{Uz}, \tag{21}$$

$$J \cdot \frac{d\omega}{dt} + T_F(\omega) - \frac{1}{2}\mathbf{I}^T \cdot \frac{\partial \mathbf{L}(\varphi)}{\partial \varphi} \cdot \mathbf{I} = T_L, \tag{22}$$

The vectors of currents and voltages are of the 6th order and it refer to concatenated three-phase stator and rotor vectors. So:

$$\mathbf{I} = \begin{bmatrix} \mathbf{Is} \\ \mathbf{Ir} \end{bmatrix}; \ \mathbf{Uo} = \begin{bmatrix} U_{os} \cdot 1_{3x1} \\ U_{or} \cdot 1_{3x1} \end{bmatrix}; \ , \tag{23}$$

where: $\mathbf{Is} = [I_{s1}\ I_{s2}\ I_{s3}]^T$, $\mathbf{Ir} = [I_{r1}\ I_{r2}\ I_{r3}]^T$, $\mathbf{1}_{3x1} = [1, 1, 1]^T$, $U_{os}$, denote voltage between center points of star power supply and star winding for the stator, $U_{or}$—voltage between center points of star winding of the rotor and the star load or power supply of rotor, **Us, Ur**—are vectors of the phase voltage that supply power to the stator and rotor windings. The motor inductance matrix $\mathbf{L}(\varphi)$ and the resistance matrix **R** are described as (15) and (16) respectively.

The voltages supplying the stator are assumed in form:

$$\mathbf{Us} = \mathbf{Es} \cdot \mathbf{vS}, \tag{24}$$

where: **Es**—the diagonal matrix of magnitudes and time functions of power supply phase voltages.

$$\mathbf{Es} = \begin{bmatrix} E_{s1} & 0 & 0 \\ 0 & E_{s2} & 0 \\ 0 & 0 & E_{s3} \end{bmatrix}; \ \mathbf{vS} = \begin{bmatrix} \sin(\omega_s t) \\ \sin(\omega_s t + q) \\ \sin(\omega_s t + 2q) \end{bmatrix}, \tag{25}$$

where: $q = 2 \cdot \pi / 3$.

The mean value of the phase voltages amplitudes is described as:

$$Es = (E_{s1} + E_{s2} + E_{s3})/3, \tag{26}$$

The (21) components are column vectors of voltages of the 6th order. Their first three rows represent the stator equation, the other three are the rotor one.

In order to simplify the writing to the dimensionless form, the voltage and currents of the rotor are transferred to the stator level. For this purpose, the stator equations are divided by the mean amplitude of the phase supply voltages $Es$, the rotor equations are divided by $Es \cdot (M/L_{sm})$. Next the time scaling $\tau = \omega_s \cdot t$ is used and then the substitution of the value of stator and rotor currents divided by $Es/(\omega_s L_{sm})$ and $Es/(\omega_s M)$ respectively is performed. After the dimensionless variables' definition:

$$\begin{aligned} \mathbf{i_s} &= \mathbf{I_s}/(Es/\omega_s L_{sm}), & \mathbf{i_r} &= \mathbf{I_r}/(Es/(\omega_s M)) \\ \mathbf{r_s} &= \mathbf{Rs}/(\omega_s L_{sm}), & \mathbf{r_r} &= \mathbf{Rr}/(\omega_s L_{rm}) \\ \mathbf{u_{os}} &= \mathbf{U_{os}}/Es, & \mathbf{u_{or}} &= \mathbf{U_{or}}/(Es \cdot (M/L_{sm})) \\ \mathbf{e_s} &= \mathbf{Es}/Es, & \mathbf{u_r} &= \mathbf{Ur}/(Es \cdot (M/L_{sm})) \end{aligned} \tag{27}$$

(21) may be written in form.

$$\mathbf{\Lambda} \cdot \begin{bmatrix} \overset{\circ}{\mathbf{i_s}} \\ \overset{\circ}{\mathbf{i_r}} \end{bmatrix} + \mathbf{\Theta} \cdot \begin{bmatrix} \mathbf{i_s} \\ \mathbf{i_r} \end{bmatrix} + \begin{bmatrix} \mathbf{u_{os}} \\ \mathbf{u_{or}} \end{bmatrix} = \begin{bmatrix} \mathbf{e_s} \cdot \mathbf{vS}(\tau) \\ \mathbf{u_r} \end{bmatrix}, \tag{28}$$

where:

$$\Lambda = \begin{bmatrix} \mathbf{mLs} & \mathbf{mC} \\ \mathbf{mC}^T & \mathbf{mLr} \end{bmatrix}, \; \Theta = \begin{bmatrix} \mathbf{r_s} & \frac{d\mathbf{mC}}{d\varphi}\omega_s \cdot \overset{\circ}{\varphi} \\ \frac{d\mathbf{mC}^T}{d\varphi}\omega_s \cdot \overset{\circ}{\varphi} & \mathbf{r_r} \end{bmatrix}, \tag{29}$$

A circle above the state variables denotes the time derivative calculated in relation to $\tau$, which describes the time after time scaling.

The three-phase equation of a stator and a rotor may be described by two instantaneous values of currents. It means that the stator and rotor circuit may be also described by two equation system of the second order. Therefore, the stator (or/and rotor) equations of the 1st and 2nd phase will be used only for the analysis. The 3rd phase current is replaced with a negative sum of the current of the first and second phases. The replacing process for the stator and rotor currents is shown as $\mathbf{T_{32}}$ transformation [14].

$$\mathbf{i_s} = \mathbf{T_{32}} \cdot \mathbf{i_{s2}}, \; \mathbf{i_r} = \mathbf{T_{32}} \cdot \mathbf{i_{r2}}, \tag{30}$$

where:

$$\mathbf{T_{32}} = \begin{bmatrix} 1 & 0 \\ 0 & 1 \\ -1 & -1 \end{bmatrix}, \tag{31}$$

In order to eliminate voltages between center points of stars of power supply and stator windings, the third phase equation should be subtracted from the first-phase equation and from the second-phase equation of stator. The same should be done with the equations of the rotor. The above subtractions correspond to a premultiplication by a matrix [16] separately for stator and rotor equations:

$$\mathbf{T_{23}} = \begin{bmatrix} 1 & 0 & -1 \\ 0 & 1 & -1 \end{bmatrix}, \tag{32}$$

After transformations the equations of the electric part of the motor may be expressed as follows:

$$\Lambda_2 \cdot \begin{bmatrix} \overset{\circ}{\mathbf{i_{s2}}} \\ \overset{\circ}{\mathbf{i_{r2}}} \end{bmatrix} + \Theta_2 \cdot \begin{bmatrix} \mathbf{i_{s2}} \\ \mathbf{i_{r2}} \end{bmatrix} = \begin{bmatrix} \mathbf{T_{23}} \cdot \mathbf{e_s} \cdot \mathbf{vS}(\tau) \\ \mathbf{T_{23}} \cdot \mathbf{u_r} \end{bmatrix}, \tag{33}$$

where:

$$\Lambda_2 = \begin{bmatrix} \mathbf{T_{23}} \cdot \mathbf{mLs} \cdot \mathbf{T_{32}} & \mathbf{T_{23}} \cdot \mathbf{mC} \cdot \mathbf{T_{32}} \\ \mathbf{T_{23}} \cdot \mathbf{mC}^T \cdot \mathbf{T_{32}} & \mathbf{T_{23}} \cdot \mathbf{mLr} \cdot \mathbf{T_{32}} \end{bmatrix} = \begin{bmatrix} \lambda_s \mathbf{mT_2} & \mathbf{mC_2} \\ \mathbf{mC_2}^T & \lambda_r \mathbf{mT_2} \end{bmatrix}, \tag{34}$$

$$\Theta_2 = \begin{bmatrix} \mathbf{T_{23}} \cdot \mathbf{r_s} \cdot \mathbf{T_{32}} & \mathbf{T_{23}} \cdot \left(\frac{d\mathbf{mC}}{d\varphi}\right) \cdot \mathbf{T_{32}} \cdot \omega_s \cdot \overset{\circ}{\varphi} \\ \mathbf{T_{23}} \cdot \left(\frac{d\mathbf{mC}^T}{d\varphi}\right) \cdot \mathbf{T_{32}} \cdot \omega_s \cdot \overset{\circ}{\varphi} & \mathbf{T_{23}} \cdot \mathbf{r_r} \cdot \mathbf{T_{32}} \end{bmatrix} = \begin{bmatrix} \mathbf{mr_{s2}} & \frac{d\mathbf{mC_2}}{d\varphi} \cdot \omega_s \cdot \overset{\circ}{\varphi} \\ \left(\frac{d\mathbf{mC_2}}{d\varphi}\right)^T \cdot \omega_s \cdot \overset{\circ}{\varphi} & \mathbf{mr_{r2}} \end{bmatrix}, \tag{35}$$

The $\Lambda$ and $\Theta$ matrices are sized $6 \times 6$ dimensions. After applying the $\mathbf{T_{23}}$ and $\mathbf{T_{32}}$ transformations, the $\Lambda_2$ and $\Theta_2$ matrices are $4 \times 4$ dimensions and their elements—submatrices are $2 \times 2$ dimensions. The values and designations of these sub-arrays for $\Lambda_2$ are as follows:

$$\mathbf{mC_2} = \mathbf{T_{23}} \cdot \mathbf{mC} \cdot \mathbf{T_{32}} = 3 \cdot \begin{bmatrix} \cos(3\varphi) & -\cos(3\varphi + 2 \cdot q) \\ -\cos(3\varphi + q) & \cos(3\varphi) \end{bmatrix}, \tag{36}$$

$$\lambda_s \cdot \mathbf{mT_2} = \mathbf{T_{23}} \cdot \mathbf{mLs} \cdot \mathbf{T_{32}}, \tag{37}$$

$$\lambda_r \cdot \mathbf{mT_2} = \mathbf{T_{23}} \cdot \mathbf{mLr} \cdot \mathbf{T_{32}}, \tag{38}$$

where:

$$\mathbf{mT_2} = \mathbf{T_{23}} \cdot \mathbf{T_{32}} = \begin{bmatrix} 2 & 1 \\ 1 & 2 \end{bmatrix}, \tag{39}$$

$$\lambda_s = 3 + 2\sigma_s, \; \lambda_r = 3 + 2\sigma_r, \tag{40}$$

Similarly, elements of the $\boldsymbol{\Theta_2}$ matrix are in the form:

$$\mathbf{mr_{s2}} = \mathbf{T}_{23} \cdot \mathbf{r_s} \cdot \mathbf{T}_{32} = \begin{bmatrix} r_{s1} + r_{s3} & r_{s3} \\ r_{s3} & r_{s2} + r_{s3} \end{bmatrix}, \tag{41}$$

$$\mathbf{mr_{r2}} = \mathbf{T}_{23} \cdot \mathbf{r_r} \cdot \mathbf{T}_{32} = \begin{bmatrix} r_{r1} + r_{r3} & r_{r3} \\ r_{r3} & r_{r2} + r_{r3} \end{bmatrix}, \tag{42}$$

$$\frac{\partial \mathbf{mC_2}}{\partial \varphi} = \mathbf{T}_{23} \cdot \frac{\partial \mathbf{mC}}{\partial \varphi} \cdot \mathbf{T}_{32} = 9 \cdot \begin{bmatrix} -\sin(3\varphi) & \sin(3\varphi + 2 \cdot q) \\ \sin(3\varphi + q) & -\sin(3\varphi) \end{bmatrix}, \tag{43}$$

Computing elements of matrices (34) and (35) yields possibility calculation of currents derivatives using equation:

$$\begin{bmatrix} \overset{\circ}{\mathbf{i_{s2}}} \\ \overset{\circ}{\mathbf{i_{r2}}} \end{bmatrix} = \boldsymbol{\Lambda_2}^{-1} \cdot \left[ \begin{bmatrix} \mathbf{T}_{23} \cdot \mathbf{e_s} \cdot \mathbf{vS}(\tau) \\ \mathbf{T}_{23} \cdot \mathbf{u_r} \end{bmatrix} - \boldsymbol{\Theta_2} \cdot \begin{bmatrix} \mathbf{i_{s2}} \\ \mathbf{i_{r2}} \end{bmatrix} \right], \tag{44}$$

The $\boldsymbol{\Lambda_2}^{-1}$—inverted matrix may be determined using the procedures presented in Appendices A and B.

The mechanical Equation (22) of the motor may also be transformed. After substitution of wet friction:

$$T_F = k_F \cdot \frac{d\varphi}{dt}, \tag{45}$$

and time scaling, it has the following form:

$$\omega_s{}^2 \cdot J \cdot \frac{d^2\varphi}{d\tau^2} + \omega_s \cdot k_F \cdot \frac{d\varphi}{d\tau} + T_L = ke \cdot \mathbf{i_{s2}}^T \cdot \frac{d\mathbf{mC_2}}{d\varphi} \cdot \mathbf{i_{r2}}, \tag{46}$$

where:

$$ke = \frac{Es^2}{\omega_s{}^2 \cdot L_{sm}}, \tag{47}$$

The mathematical model describes the inductive slip-ring motor with symmetrical inductances. The remaining elements of the model may be asymmetric, i.e., they may have unequal phase components. The tests conducted on the model may be divided into the analysis of the model of the symmetrical arrangement and sensitivity studies of the remaining elements affecting its characteristics. The model uses dimensionless variables and parameters and that is why it is simple and maybe useful in simulation of the slip-ring induction motor.

## 3. The Model of the Motor in Simulink

The motor electric and mechanical Equations (44) and (46) respectively, provide the basis for the creation of the vectorized simulation model in Simulink. The vectorization simplifies the diagram of the model and facilitates its use. The model diagram is presented in Figure 3.

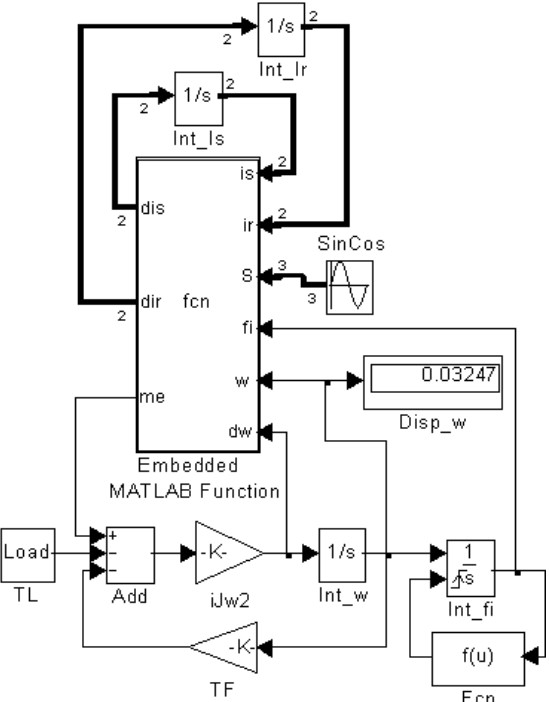

**Figure 3.** The model of a slip-ring motor in a Simulink program.

The physical parameters of the modeled motor are presented in Table 1.

**Table 1.** Parameters of the modeled induction motor.

| Name | Value | Name | Value |
|------|-------|------|-------|
| $E_s$ | 230·sqrt(2) V | $L_{rm}$ | 3.9 mH |
| $\omega_s$ | 2·50·$\pi$ rad/s | $L_{r\_}$ | 0.55 mH |
| $R_s$ | 10.5 $\Omega$ | $R_r$ | 0.523 $\Omega$ |
| $L_{s\_}$ | 0.0293 H | $q$ | 2·$\pi$/3 rad |
| $L_{sm}$ | 0.187 H | $kT$ | 0.005 Nm·s |
| $M$ | 0.027 H | $J$ | 0.011 kg·m$^2$ |

Physical variables are converted into dimensionless Simulink input variables and simulation output variables into physical variables of the object in the MATLAB script, which controls the simulation experiment.

The electric currents and their time derivatives are used in the model in the form of two-dimensional signal vectors. These signals lines are presented in the diagram as thick lines. The equations of the electrical part are integrated in the upper part of the model diagram. The mechanical part is solved in the lower part of the diagram. In the middle there is a block of the built-in MATLAB function, in which the equations of the state co-ordinates derivatives are formulated. These derivatives for the electrical part are determined from (44) and for the mechanical part from (46) from dimensionless variables. The simulation was performed under zero initial conditions. The results of the simulation are placed in MATLAB workspace. Then they are used to draw up the waveforms of the model.

The diagram of the model is very compact and clear. The return to physical quantities of the model variables is performed after the simulation. The rescaling calculations have been completed and then the figures are drawn.

The simplicity of the final Simulink model ought to be emphasized. It results from the application of the transformation of the circuit equation to (44) and the use of vector-matrix notation. The simplicity

of the model may be estimated by comparing it with the models presented in [4,13,17]. The obtained model is also more convenient for the arrangement of simulation experiments.

## 4. T-Equivalent Induction Motor Model

For a rough check of the modelling results, it was decided to use the possibly simple model of an induction motor. It was decided to use the Steinmetz model. His replacement diagram of a multi-phase induction motor in the form of T-equivalent circuit is presented in Figure 4.

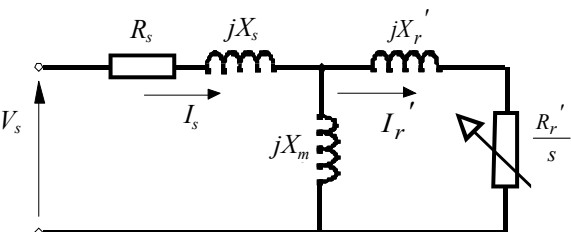

**Figure 4.** Steinmetz T-equivalent diagram of induction motor.

The following components are used in the diagram:

- $R_s$, $X_s$—stator resistance and leakage reactance of stator,
- $X_m$—motor magnetizing reactance,
- $R_r'$, $X_r'$—rotor resistance and leakage reactance of rotor transformed to stator side,
- $s$—slip.

This is a single-phase model of a multiphase induction motor. The model of the motor is valid in steady-state balanced circuit condition [6]. The slip of the motor is defined as

$$s = \frac{\omega_s - 3\omega_r}{\omega_s}, \tag{48}$$

Synchronous angular velocities of the stator's magnetic field rotation and the rotor's angular speed were determined as $\omega_s$, $\omega_r$ respectively. The element on which the output load is generated in this scheme is the resistance with the value $R_r'/s$. Steinmetz presented the power emitted on this resistance as a sum of electromechanical output power and thermal power. To determine these powers, he introduced the following equation:

$$\frac{R_r'}{s} = R_r' \cdot \frac{1-s}{s} + R_r', \tag{49}$$

Multiplying this equation by $3 \cdot I_r'^{\,2}$ the following equation is obtained

$$P_{gap} = P_{em} + P_r, \tag{50}$$

in which the $P_{gap}$ is air gap power, $P_{em}$—electromechanical output power and $P_r$—the heat power generated on the resistance. The above active power of rotor is respectively equal to:

$$P_{gap} = 3I_r'^2 \cdot R_r' \frac{1}{s} = 3I_r'^2 \cdot R_r' \cdot \frac{\omega_s}{\omega_s - 3\omega_r}, \tag{51}$$

$$P_{em} = 3I_r'^2 \cdot R_r' \cdot \frac{1-s}{s} = 3I_r'^2 \cdot R_r' \cdot \frac{\omega_r}{\omega_s/3 - \omega_r}, \tag{52}$$

$$P_r = 3I_r'^2 \cdot R_r', \tag{53}$$

To simplify further analysis, the IEEE recommends using Thevenin's claim. The result is the diagram shown in Figure 5.

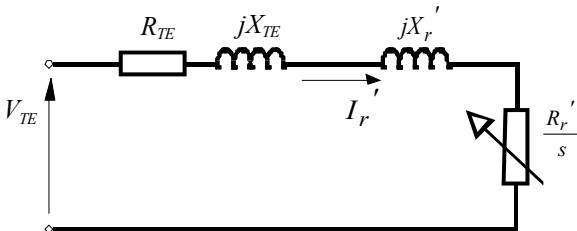

**Figure 5.** Steinmetz T-equivalent diagram of induction motor after using Thevenin's Theorem.

In this diagram there is a voltage source with $V_{TE}$ voltage and $Z_{TE}$ impedance in series with rotor reactance and load resistance. These variables are equal to:

$$V_{TE} = \frac{jX_m}{Z_s + jX_m} V_s \qquad Z_{TE} = \frac{jX_m \cdot Z_s}{Z_s + jX_m}, \tag{54}$$

where $Z_s = R_s + jX_s$.

The current $I_r'$ flowing in the load may be calculated as follows:

$$I_r' = \frac{V_{TE}}{Z_{TE} + jX_r' + \frac{R_r'}{s}} = \frac{V_s}{Z_s + \left(jX_r' + \frac{R_r'}{s}\right) \cdot \left(1 + \frac{Z_s}{jX_m}\right)}, \tag{55}$$

Using the above result, it is possible to determine the stator current from the circuit in Figure 4:

$$I_s = \left(1 + \frac{X_r'}{X_m} - j\frac{1}{X_m}\frac{R_r'}{s}\right)I_r', \tag{56}$$

The electromechanical output power can be determined from (52) and on this basis the electromechanical moment may be calculated.

$$T_{em} = \frac{P_{em}}{\omega_r}, \tag{57}$$

It should be stressed that the above dependencies result from the scheme adopted by Steinmetz and the assumption that the individual phase circuits of a three-phase induction motor can be treated as working independently on torque.

## 5. Example of the Model Waveforms

The results of the simulation were compared with the values determined for the same motor parameters using the Steinmetz model, which allows us to determine the motor currents and powers. However, the full state vector in this model cannot be reproduced as the rotor speed is not available. The simulation model is used to determine the steady state speed as a function of the load torque. The course of this speed during motor start-up for the load torque $T_L = 1$ Nm and $T_L = 15$ Nm is shown in Figure 6. These graphs show that after just one period of run there is an increase in motor speed equal to about 20% of steady state speed. The time of reaching the steady state is equal to about 5 to 10 periods of supply voltage. The slip for the load torque $T_L = 1$ Nm is about 0.025 and for $T_L = 15$ Nm it is 0.435.

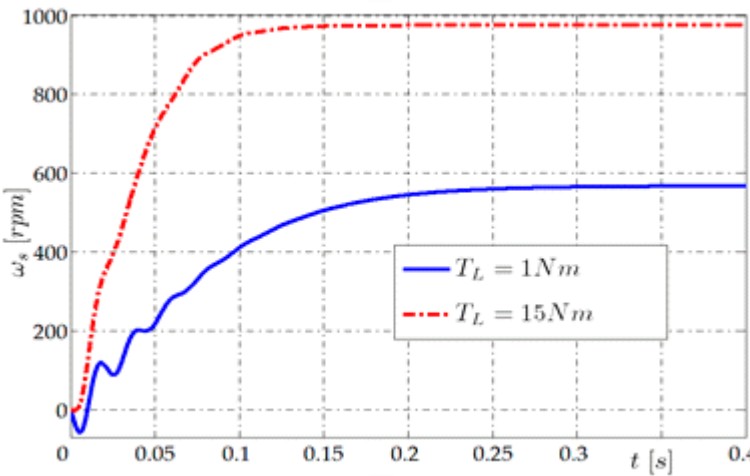

**Figure 6.** The rotation velocity during start-up process of the motor for different load torque values.

The dynamics of the start-up process are visible on the graphs of rotor currents—Figures 7 and 8 and stator—Figures 9 and 10. Figure 7 shows the rotor currents for load torque $T_L = 1$ Nm. In steady state they had an amplitude equal to 2 A and a period of 0.82 s which corresponds to a slip equal to 0.025. The current amplitude at the beginning of the start-up was 48 A.

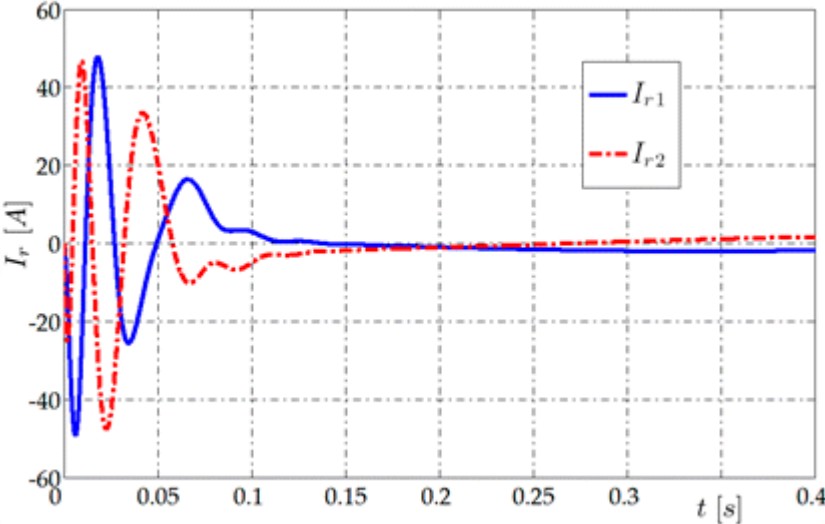

**Figure 7.** The phase currents in the rotor windings during the start-up motor for load torque 1 Nm.

In Figure 8, for torque $T_L = 15$ Nm in steady state it was about 5.5 A and the period was about 0.046 s. At the beginning of the start-up process, the current amplitude reached 50 A. The determined periods of current oscillation of the rotor are in accordance with the rotor speed courses in Figure 6. The frequency and the magnitude of the currents waveforms of the rotor decreases with the reduction in the load torque.

Starting currents can be estimated using Steinmetz model dependencies. The observed model data differs from those calculated for $s = 1$ using (55) by about 5 percent of the later ones. This can be explained by a fairly high acceleration during the first period and a reduction in slip s, which also reduces the rotor current.

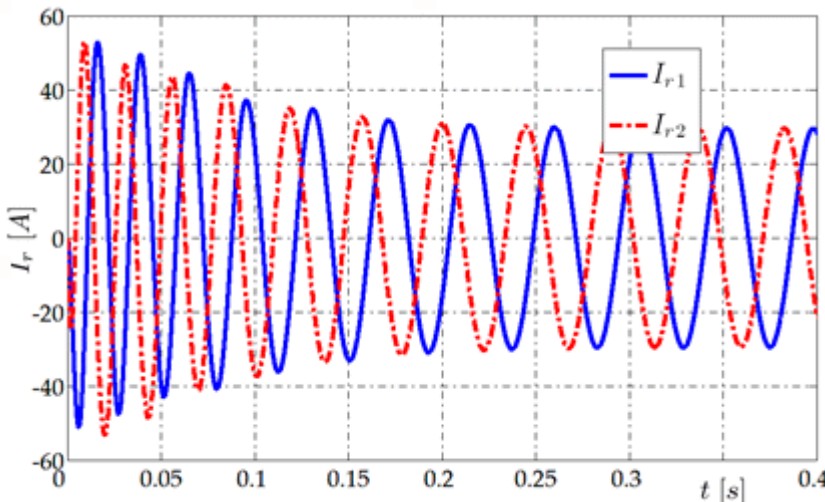

**Figure 8.** The phase currents in the rotor windings during the start-up motor for load torque 15 Nm.

From the rotor current, it is possible to determine the value of the stator current amplitude at the beginning of the motor start-up process for both load torque values. These currents are shown in Figures 9 and 10.

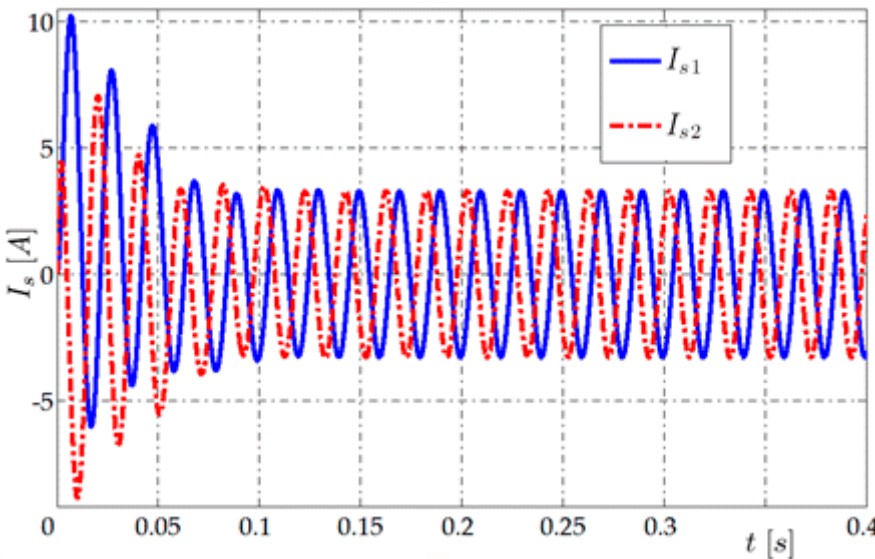

**Figure 9.** The phase currents in the stator windings during the start-up motor for load torque 1 Nm.

The initial value (for $s = 1$) determined from relation (56) is about 10.4 A as in these figures. Differences occur for steady state stator current amplitudes. For $T_L = 1$ Nm the calculation shows amplitude of 4.7 A and the simulation in Figure 9 shows a value of about 3.3 A. Similarly, for $T_L = 15$ Nm the calculation gives 6.2 A and the graph in Figure 10 shows 5.5 A.

The stator current waveforms presented in Figures 9 and 10, have two stages: the start-up and steady-state. The shape of the waveforms during the stages is clear. In the case of a smaller load torque the current stabilizes after only four periods of the power supply voltage. The magnitudes of currents in steady-state rise with the increase in load torque value. The stator currents waveforms frequency is independent of the load torque and constant, and equal to the frequency of the supply network.

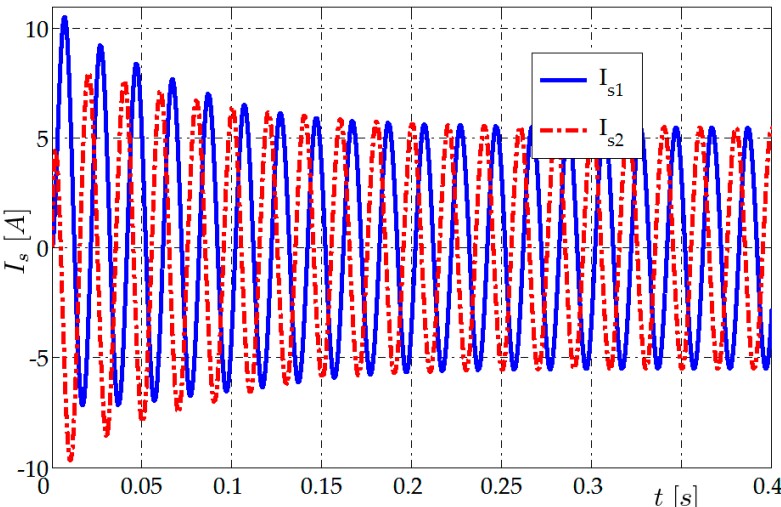

**Figure 10.** The phase currents in the stator windings during the start-up motor for load torque 15 Nm.

The electric torque during the start-up motor for the different load torque values is presented in Figure 11.

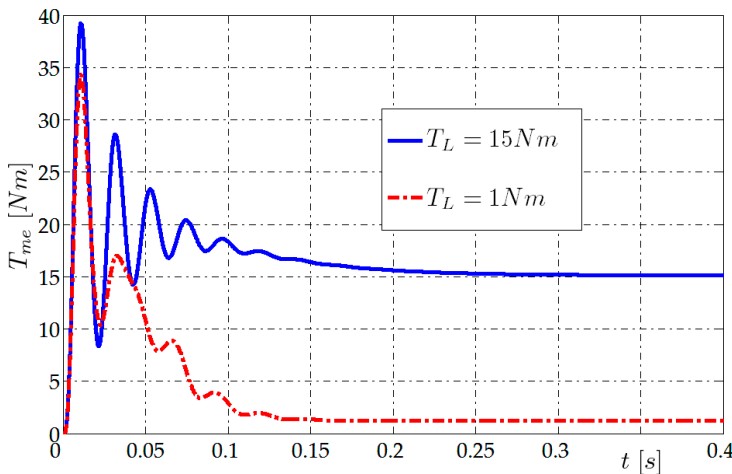

**Figure 11.** The electrical torque during the start-up motor for different load torque.

In the steady state, the electromagnetic torque waveforms are close to the load torque. This is due to the inclusion of the mechanical part in the simulation model. The Steinmetz model for load torque $T_L = 1$ Nm, gives torque value approximately 30% lower than the simulation steady state. The compliance was obtained for the higher value of the load moment. It seems that it may result from the simplification of the T-equivalent model. The T-model is sensitive for the slip value close to zero. In both cases the constant component of the start-up torque appears. It determines the dynamics of the start-up process of the motor. After reaching a specified angular speed, the electric torque converges to the sum of torques of loads and friction.

## 6. Conclusions

The model of slip-ring induction motor presented in this paper was obtained on the basis of inductance matrix in the form specified in the literature. The elements of this matrix were determined experimentally for the selected motor using methods developed by the authors. After the Lagrange function was formulated, motion equations of the motor were determined, dimensionless variables were introduced and the minimal form of motion equations of the electric part of the induction motor

model were established. These equations are recorded in the vector- matrix form of the fourth order. These are non-linear equations, with non-linearity resulting from the dependence of the inductance matrix on the angle of rotor rotation.

These equations are original. The form of these equations allowed to formulate a simple operational diagram of the slip-ring induction motor model in Simulink. In order to check the correctness of the model, the T-equivalent diagram of the induction motor was presented. The correspondence between the initial period of the motor start processes and some variables of the steady state vector was obtained. Such possibilities were made possible using this additional model.

Three-phase circuits of the motor stator and rotor are described using differential equations of two-phase currents of the stator and of two-phase currents of the rotor circuits. As a result, the motor circuits are more easily observed and the forms of the equations are simpler.

The model may be easily adapted for the research on the influence of asymmetry of power supply, dynamical load of the motor and influence of unbalance of other components of motor circuits. The preliminary results of tests are promising. The tests conducted on the model may be divided into the analysis of the model of the balanced (symmetrical) arrangement and sensitivity studies of the remaining elements affecting its characteristics as in [14].

The model makes simulation of a slip-ring induction motor simpler than the ones in [9,13,17]. The authors consider a concept of the presented mathematical transformations and the model as a novelty.

**Author Contributions:** Conceptualization, M.W. and K.S.; methodology, M.W.; software, K.S.; validation, M.W., K.S. and A.C.; formal analysis, M.W.; investigation, K.S. and A.C.; resources, K.S. and A.C.; data curation, K.S.; writing—original draft preparation, K.S. and A.C.; writing—review and editing, M.W. and A.C.; visualization, K.S.; supervision, M.W.; project administration, M.W.; funding acquisition, M.W. and A.C. All authors have read and agreed to the published version of the manuscript.

**Funding:** This research received no external funding.

**Conflicts of Interest:** The authors declare no conflict of interest.

## Appendix A. Algorithm of Matrix Inversion

A matrix of 2n × 2n dimension is considered in the form given by:

$$\begin{bmatrix} \mathbf{A} & \mathbf{B} \\ \mathbf{C} & \mathbf{D} \end{bmatrix}, \tag{A1}$$

**A**, **B**, **C**, **D** denote submatrices of n × n dimension. It is assumed that inverse matrices of submatrices **A**, **D** exist and that the inverse matrix of the (A1) may be written in the same form as the input matrix. Submatrices of the inverse are denoted with additional letter **i**. The inverse matrix and the original matrix fulfill the relation:

$$\begin{bmatrix} \mathbf{Ai} & \mathbf{Bi} \\ \mathbf{Ci} & \mathbf{Di} \end{bmatrix} \cdot \begin{bmatrix} \mathbf{A} & \mathbf{B} \\ \mathbf{C} & \mathbf{D} \end{bmatrix} = \begin{bmatrix} 1_{Dn} & 0_{nxn} \\ 0_{nxn} & 1_{Dn} \end{bmatrix}, \tag{A2}$$

where: $\mathbf{1}_{Dn}$ marks the identity matrix of n x n dimensions, and $\mathbf{0}_{nxn}$ is the zero matrix of n x n dimensions.

After the multiplication of the submatrices the following equations are obtained:

$$\mathbf{Ai} \cdot \mathbf{A} + \mathbf{Bi} \cdot \mathbf{C} = 1_{Dn}, \tag{A3}$$

$$\mathbf{Ai} \cdot \mathbf{B} + \mathbf{Bi} \cdot \mathbf{D} = 0_{nxn}, \tag{A4}$$

$$\mathbf{Ci} \cdot \mathbf{A} + \mathbf{Di} \cdot \mathbf{C} = 0_{nxn}, \tag{A5}$$

$$\mathbf{Ci} \cdot \mathbf{B} + \mathbf{Di} \cdot \mathbf{D} = 1_{Dn}, \tag{A6}$$

From (A4) we get:

$$\mathbf{Bi} = -\mathbf{Ai} \cdot \mathbf{B} \cdot \mathbf{D}^{-1}, \tag{A7}$$

and after substituting into (A3):

$$\mathbf{Ai} \cdot \mathbf{A} - \mathbf{Ai} \cdot \mathbf{B} \cdot \mathbf{D}^{-1} \cdot \mathbf{C} = 1_{Dn}, \tag{A8}$$

Hence

$$\mathbf{Ai} = \left(\mathbf{A} - \mathbf{B} \cdot \mathbf{D}^{-1} \cdot \mathbf{C}\right)^{-1} = \left(1_{Dn} - \mathbf{A}^{-1} \cdot \mathbf{B} \cdot \mathbf{D}^{-1} \cdot \mathbf{C}\right)^{-1} \cdot \mathbf{A}^{-1}, \tag{A9}$$

The substitution of the above into (A7) yields:

$$\mathbf{Bi} = -\left(1_{Dn} - \mathbf{A}^{-1} \cdot \mathbf{B} \cdot \mathbf{D}^{-1} \cdot \mathbf{C}\right)^{-1} \cdot \mathbf{A}^{-1} \cdot \mathbf{B} \cdot \mathbf{D}^{-1}, \tag{A10}$$

Similarly, the remaining submatrices can be obtained as follows:

$$\mathbf{Ci} = -\left(1_{Dn} - \mathbf{D}^{-1} \cdot \mathbf{C} \cdot \mathbf{A}^{-1} \cdot \mathbf{B}\right)^{-1} \cdot \mathbf{D}^{-1} \cdot \mathbf{C} \cdot \mathbf{A}^{-1}, \tag{A11}$$

$$\mathbf{Di} = \left(\mathbf{D} - \mathbf{C} \cdot \mathbf{A}^{-1} \cdot \mathbf{B}\right)^{-1} = \left(1_{Dn} - \mathbf{D}^{-1} \cdot \mathbf{C} \cdot \mathbf{A}^{-1} \cdot \mathbf{B}\right)^{-1} \cdot \mathbf{D}^{-1}, \tag{A12}$$

## Appendix B. Algorithm of the Inversion of Inductance Matrix

The analysis of the matrix inversion in (44) may be facilitated if cosine functions in (12) will be replaced by:

$$a = \cos(3\varphi) \quad b = \cos(3\varphi + q) \quad c = \cos(3\varphi + 2 \cdot q) \quad q = 2 \cdot \pi/3, \tag{A13}$$

Then the (12) has the form:

$$\mathbf{mC} = \begin{bmatrix} a & b & c \\ c & a & b \\ b & c & a \end{bmatrix}, \tag{A14}$$

and its elements fulfill the relation

$$a + b + c = 0, \tag{A15}$$

It is easy to obtain the equality (36) in form:

$$\mathbf{mC_2} = \mathbf{T_{3w2}} \cdot \mathbf{mC} \cdot \mathbf{T_{2w3}} = 3 \begin{bmatrix} a & -c \\ -b & a \end{bmatrix}, \tag{A16}$$

In (44) the inversion of the matrix $\mathbf{\Lambda_2}$ (34), which submatrices are denoted as follows:

$$\begin{bmatrix} \lambda_s \cdot \mathbf{mT_2} & \mathbf{mC_2} \\ \mathbf{mC_2}^T & \lambda_r \cdot \mathbf{mT_2} \end{bmatrix} = \begin{bmatrix} \mathbf{A} & \mathbf{B} \\ \mathbf{C} & \mathbf{D} \end{bmatrix}, \tag{A17}$$

The matrices **A, D** are proportional to the $\mathbf{mT_2}$, which may be easily inverted:

$$\mathbf{mT_2}^{-1} = \frac{1}{3} \begin{bmatrix} 2 & -1 \\ -1 & 2 \end{bmatrix}, \tag{A18}$$

In order to obtain the inverse matrix of (A17), the matrix products $\mathbf{A^{-1} \cdot B}$ and $\mathbf{D^{-1} \cdot C}$ from—Appendix A will be helpful. After substituting equations (A16), (A18) and taking into account the relation (A15) these products may be presented in the form:

$$\mathbf{A}^{-1} \cdot \mathbf{B} = \lambda_s^{-1} \mathbf{mT_2}^{-1} \cdot \mathbf{mC_2} = \lambda_s^{-1} \cdot \frac{1}{3} \begin{bmatrix} 2 & -1 \\ -1 & 2 \end{bmatrix} \cdot 3 \begin{bmatrix} a & -c \\ -b & a \end{bmatrix} = \lambda_s^{-1} \begin{bmatrix} a-c & b-c \\ c-b & a-b \end{bmatrix}, \tag{A19}$$

$$\mathbf{D}^{-1} \cdot \mathbf{C} = \lambda_r^{-1} \mathbf{m} \mathbf{T}_2^{-1} \cdot \mathbf{m} \mathbf{C}_2{}^T = \lambda_r^{-1} \cdot \frac{1}{3} \begin{bmatrix} 2 & -1 \\ -1 & 2 \end{bmatrix} \cdot 3 \begin{bmatrix} a & -b \\ -c & a \end{bmatrix} = \lambda_r^{-1} \begin{bmatrix} a-b & c-b \\ b-c & a-c \end{bmatrix}, \tag{A20}$$

In all submatrices, the product $\mathbf{D}^{-1} \cdot \mathbf{C} \cdot \mathbf{A}^{-1} \cdot \mathbf{B}$ appears, which may be written as follows:

$$\mathbf{D}^{-1} \cdot \mathbf{C} \cdot \mathbf{A}^{-1} \cdot \mathbf{B} = \lambda_s{}^{-1} \cdot \lambda_r{}^{-1} \cdot \mathbf{m} \mathbf{T}_2{}^{-1} \cdot \mathbf{m} \mathbf{C}_2{}^T \cdot \mathbf{m} \mathbf{T}_2{}^{-1} \cdot \mathbf{m} \mathbf{C}_2, \tag{A21}$$

After substituting (A19), (A20), (A21) we obtain:

$$\mathbf{A}^{-1} \cdot \mathbf{B} \cdot \mathbf{D}^{-1} \cdot \mathbf{C} = \mathbf{D}^{-1} \cdot \mathbf{C} \cdot \mathbf{A}^{-1} \cdot \mathbf{B} = \cdot \lambda_s{}^{-1} \cdot \lambda_r{}^{-1} \cdot \left( (a-c)(a-b) + (b-c)^2 \right) \cdot \begin{bmatrix} 1 & 0 \\ 0 & 1 \end{bmatrix}, \tag{A22}$$

The expansion of this expression while considering (A15) allows us to obtain:

$$\mathbf{A}^{-1} \cdot \mathbf{B} \cdot \mathbf{D}^{-1} \cdot \mathbf{C} = \lambda_s{}^{-1} \cdot \lambda_r{}^{-1} \frac{3}{2} \cdot \left( a^2 + b^2 + c^2 \right) \cdot 1_{D2}, \tag{A23}$$

Using (A13) and the identities

$$\begin{aligned} a^2 &= \cos^2(3\varphi) = \frac{1}{2} + \frac{1}{2}\cos(2 \cdot 3\varphi) \\ b^2 &= \cos^2(3\varphi + q) = \frac{1}{2} + \frac{1}{2}\cos(2 \cdot 3\varphi + 2q), \\ c^2 &= \cos^2(3\varphi + 2q) = \frac{1}{2} + \frac{1}{2}\cos(2 \cdot 3\varphi + q) \end{aligned} \tag{A24}$$

The sum of squares in (A23) may be calculated as follows:

$$a^2 + b^2 + c^2 = \frac{3}{2}, \tag{A25}$$

and then the following relation is obtained:

$$\mathbf{A}^{-1} \cdot \mathbf{B} \cdot \mathbf{D}^{-1} \cdot \mathbf{C} = (9/4) \cdot \lambda_s{}^{-1} \cdot \lambda_r{}^{-1} \cdot 1_{D2}, \tag{A26}$$

Hence the submatrices of the diagonal of the inverted matrix may be written in the form:

$$\begin{aligned} \mathbf{Ai} &= \left( 1_{D2} - \mathbf{A}^{-1} \cdot \mathbf{B} \cdot \mathbf{D}^{-1} \cdot \mathbf{C} \right)^{-1} \cdot \mathbf{A}^{-1} = \left( 1 - (9/4) \cdot \lambda_s{}^{-1} \cdot \lambda_r{}^{-1} \right)^{-1} \cdot \lambda_s{}^{-1} \cdot \frac{1}{3} \begin{bmatrix} 2 & -1 \\ -1 & 2 \end{bmatrix} =, \\ &= (\lambda_s \cdot \lambda_r - 9/4)^{-1} \lambda_s \cdot \lambda_r \cdot \lambda_s{}^{-1} \cdot \frac{1}{3} \begin{bmatrix} 2 & -1 \\ -1 & 2 \end{bmatrix} = (\lambda_s \cdot \lambda_r - 9/4)^{-1} \lambda_r \cdot \frac{1}{3} \begin{bmatrix} 2 & -1 \\ -1 & 2 \end{bmatrix} \end{aligned} \tag{A27}$$

$$\mathbf{Di} = \left( 1_{Dn} - \mathbf{D}^{-1} \cdot \mathbf{C} \cdot \mathbf{A}^{-1} \cdot \mathbf{B} \right)^{-1} \cdot \mathbf{D}^{-1} = (\lambda_s \cdot \lambda_r - 9/4)^{-1} \cdot \lambda_s \cdot \frac{1}{3} \begin{bmatrix} 2 & -1 \\ -1 & 2 \end{bmatrix}, \tag{A28}$$

For the calculation of the remaining submatrices, the following products, which may be calculated from (A26) are useful:

$$\mathbf{A}^{-1} \cdot \mathbf{B} \cdot \mathbf{D}^{-1} = \lambda_s{}^{-1} \cdot \lambda_r{}^{-1} \cdot \mathbf{C}^{-1} = \lambda_s{}^{-1} \cdot \lambda_r{}^{-1} \cdot \left( \mathbf{m} \mathbf{C}_2{}^T \right)^{-1} = \lambda_s{}^{-1} \lambda_r{}^{-1} \begin{bmatrix} a & b \\ c & a \end{bmatrix}, \tag{A29}$$

$$\mathbf{D}^{-1} \cdot \mathbf{C} \cdot \mathbf{A}^{-1} = \lambda_s{}^{-1} \cdot \lambda_r{}^{-1} \cdot \mathbf{B}^{-1} = \lambda_s{}^{-1} \cdot \lambda_r{}^{-1} \cdot \mathbf{m} \mathbf{C}_2{}^{-1} = \lambda_s{}^{-1} \lambda_r{}^{-1} \begin{bmatrix} a & c \\ b & a \end{bmatrix}, \tag{A30}$$

Using (A26) and (A29) and (A30) respectively, the following elements are calculated:

$$\mathbf{Bi} = -\left( 1_{D2} - \mathbf{A}^{-1} \cdot \mathbf{B} \cdot \mathbf{D}^{-1} \cdot \mathbf{C} \right)^{-1} \cdot \mathbf{A}^{-1} \cdot \mathbf{B} \cdot \mathbf{D}^{-1} = -(\lambda_s \cdot \lambda_r - 9/4)^{-1} \cdot (9/4) \cdot \left( \mathbf{m} \mathbf{C}_2{}^T \right)^{-1}, \tag{A31}$$

$$\mathbf{Ci} = -\left( 1_{D2} - \mathbf{D}^{-1} \cdot \mathbf{C} \cdot \mathbf{A}^{-1} \cdot \mathbf{B} \right)^{-1} \cdot \mathbf{D}^{-1} \cdot \mathbf{C} \cdot \mathbf{A}^{-1} = -(\lambda_s \cdot \lambda_r - 9/4)^{-1} \cdot (9/4) \cdot \mathbf{m} \mathbf{C}_2{}^{-1}, \tag{A32}$$

The inverted matrix (A17) i.e., $\mathbf{\Lambda}_2$ (34), may be written as:

$$\begin{bmatrix} \lambda_s \cdot \mathbf{mT}_2 & \mathbf{mC}_2 \\ \mathbf{mC}_2{}^T & \lambda_r \cdot \mathbf{mT}_2 \end{bmatrix}^{-1} = (\lambda_s \cdot \lambda_r - 9/4)^{-1} \cdot \begin{bmatrix} \lambda_r \cdot \mathbf{mT}_2{}^{-1} & -(9/4) \cdot \left(\mathbf{mC}_2{}^T\right)^{-1} \\ -(9/4) \cdot \mathbf{mC}_2{}^{-1} & \lambda_s \cdot \mathbf{mT}_2{}^{-1} \end{bmatrix}, \quad \text{(A33)}$$

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
