# Peer review of "Vectorized Mathematical Model of a Slip-Ring Induction Motor†"

_energies, doi:10.3390/en13154015_

Round 1

Reviewer 1 Report

The main advantage of the paper is its serious mathematical backgroung with necessary explanation. At the same time, there are some remarks as follow:

  1. The goal and new original contribution of the paper are not clear enough in my opinion and must be emphasized/improved in more detail.
  2. The introduction should be improved and supplemented with more new actual journal articles related to this topic. 
  3. The paper lacks some table with all necessary parameters used for modeling in Simulink.
  4. The presentation of the paper is not exciting, it is not very attractive for readers in my opinion because:
    4.1 there are on;y black-white colours used;
    4.2 the main attention in the paper was devoted to explanation and formulation of the theoretical and mathematical background, but not so much for verification and interpretation of the results.

Author Response

Please find attached file of the response

Reviewer 2 Report

This paper deals with the determination of the mathematical model of the slip-ring induction motor. Although the topic is interesting, the paper does not seem to be qualified for publication in Energies due to the lack of novelty. The proposed model seems to be basic and classical. In addition, no comparative performance analysis is carried out with any of the previously presented/available mathematical models of slip-ring IM.  

Author Response

Please find attached the file of the response.

Reviewer 3 Report

This article presents a procedure for electric parameters identification for a three-phase slip-ring induction motor, and a numerical model for the motor. The identification procedure provides a simplified practical way to obtain the motor’s parameters through doable experiments. On the other hand, the article takes an interesting approaches to jointly do simulation on electrical and mechanical systems via energy equations.

The results have been produced by applying the approaches presented in the article in parameter identification and motor simulation. However, it could recommended that comparison of results with other approaches or experiments is presented for verification.

Besides some minor typos, some equation referrals are not clear, such as in line (146) equation (6) has been mentioned, which appears not clearly right. In line (205) equation (24) has been mentioned to produce equation (28), which is also not clear so.

Author Response

(The authors gave the same response as above.)

Round 2

Reviewer 1 Report

I have appreciated the authors' efforts on the improvement of the paper scientific quality. The scientific novelty and new original contributions of the paper, which were unclear, have become more clarified. In my opinion, the most significant drawback of the paper is the absence of the complete experimental verification (physical modeling). The authors have presented the simulation results however. In my opinion, the authors' results presented in this paper, including the appendices, will be of certain value for the researchers and engineers in the field.

Author Response

Dear Reviewer 1,

Thank you for both your reviews. As engineers, we are used to black-white drawings. Your review made us aware of the need to use the presentation techniques available today.

The article was written in the initial stage of our project.  Therefore, we first took measurements to determine the relationship between inductances and resistance as a function of the rotor angle to the stator. In the next stage, we focused on defining the mathematical model of the induction motor. This model would be verified. A stand is being built to enable this process and further research we mentioned in the paper. But today we only compared the results of modeling during the research with nominal data.

Thank you again.

Best regards

Reviewer 2 Report

The authors have answered all the concerns raised by the reviewer. This paper can now be published in its current form.

Author Response

Dear Reviewer 2,

We'd like to thank you for both your reviews. We had to rethink the structure, the layout of the article. First, we enjoy mathematical solutions. Your review made us realize that the presentation should be more readable and friendly for readers and potential users of induction motors.

Thank you again

Best regards